# Methodology for Addressing Infectious Aerosol Persistence in Real-Time Using Sensor Network

**DOI:** 10.3390/s21113928

**Published:** 2021-06-07

**Authors:** Sepehr Makhsous, Joelle M. Segovia, Jiayang He, Daniel Chan, Larry Lee, Igor V. Novosselov, Alexander V. Mamishev

**Affiliations:** 1Department of Mechanical Engineering, University of Washington, 3900 E Stevens Way NE, Seattle, WA 98195, USA; jsegovia@uw.edu (J.M.S.); jh846@uw.edu (J.H.); ivn@uw.edu (I.V.N.); 2School of Dentistry, University of Washington, 1959 NE Pacific St., B-307, Seattle, WA 98195, USA; dcnchan@uw.edu; 3Pacific Industrial Hygiene, LLC, 5520 106th Ave NE, Kirkland, WA 98033, USA; pacific.ih@comcast.net; 4Department of Electrical and Computer Engineering, University of Washington, Paul Allen Center, 185 E Stevens Way NE AE100R, Seattle, WA 98195, USA; mamishev@uw.edu

**Keywords:** aerosols, dental clinics, infection control, high-volume evacuation, extra-oral suction device, particle concentration, sensor network, dispersion modeling, exposure assessment, air quality

## Abstract

Human exposure to infectious aerosols results in the transmission of diseases such as influenza, tuberculosis, and COVID-19. Most dental procedures generate a significant number of aerosolized particles, increasing transmission risk in dental settings. Since the generation of aerosols in dentistry is unavoidable, many clinics have started using intervention strategies such as area-filtration units and extraoral evacuation equipment, especially under the relatively recent constraints of the pandemic. However, the effectiveness of these devices in dental operatories has not been studied. Therefore, the ability of dental personnel to efficiently position and operate such instruments is also limited. To address these challenges, we utilized a real-time sensor network for assessment of aerosol dynamics during dental restoration and cleaning producers with and without intervention. The strategies tested during the procedures were (i) local area High-Efficiency Particle Air (HEPA) filters and (ii) Extra-Oral Suction Device (EOSD). The study was conducted at the University of Washington School of Dentistry using a network of 13 fixed sensors positioned within the operatory and one wearable sensor worn by the dental operator. The sensor network provides time and space-resolved particulate matter (PM) data. Three-dimensional (3D) visualization informed aerosol persistence in the operatory. It was found that area filters did not improve the overall aerosol concentration in dental offices in a significant way. A decrease in PM concentration by an average of 16% was observed when EOSD equipment was used during the procedures. The combination of real-time sensors and 3D visualization can provide dental personnel and facility managers with actionable feedback to effectively assess aerosol transmission in medical settings and develop evidence-based intervention strategies.

## 1. Introduction

The performance of dental procedures for patients infected with SARS-CoV-2 or other infectious diseases can produce long-lived aerosols containing infectious virus particles [1,2,3,4,5]. Due to the limitation of aerosol-monitoring instruments, there is limited information on the number, size, persistence, and fate of these aerosols in medical facilities and dental offices [6]. These limited data from simulated dental treatment suggests that dental aerosols can travel more than six feet from the patient’s mouth and remain suspended in the operatory for up to 30 min, depending upon particle size. The COVID-19 pandemic highlighted that many infections could be transmitted through aerosol exposure, establishing the urgent need to monitor the environment and develop evidence-based intervention strategies.

Many infectious disease carriers show no symptoms, making it difficult for dental practices to assess the risk of each patient [7]. To help mitigate the risk of transmission, the Center For Disease Control (CDC) recommended that vacuum and filtration units be used during and immediately after patient procedures [8]. These mitigation strategies can help minimize the risk of airborne disease transmissions by removing aerosolized particles using intervention techniques, such as High-Efficiency Particle Air (HEPA) filtration units and an Extra-Oral Suction Device (EOSD). However, there is currently limited information on how effective these interventions are and how quickly they can bring the air quality back to its original levels to ensure a safe operatory environment. If the intervention strategy fails to eliminate aerosol transmission, infectious particles can be spread via three different paths, as demonstrated in Figure 1.

Effective aerosol monitoring should enable dental clinics to assess the efficiency of the existing intervention strategies. However, professional air-monitoring units can cost thousands of dollars per unit and are typically available only in research labs or from air-monitor consulting services. Typical reference instruments do not have networking capabilities or over-air data transfer. These factors make air-quality monitoring impractical for many clinics for continuous air monitoring. This paper presents a real-time, low-cost, and comprehensible air monitoring system called AeroSpec. This scalable sensor network approach utilizes Optical Particle Counters (OPCs) to gather time and space-resolved aerosol concentration and particle-size distribution data. It is capable of transmitting information from an individual sensor via Internet of Things (IoT) protocol, displaying the data in a custom user portal, and alerting users of high concentration levels.

Low-cost particulate matter (PM) sensors find increasing use in various applications, including monitoring air quality (AQ) in the outdoor [9,10,11,12] and indoor environment [13]. Time-resolved exposure data from wearable monitors [14] can be used to assess individual exposure in near real-time. As low-cost sensors find applications in pollution monitoring, there is a need to characterize and calibrate their performance under various conditions, and especially in controlled environments with standardized test aerosols. Various studies have evaluated the performance of low-cost PM sensors in laboratory and field settings [15,16,17,18,19,20,21,22]; these reports show that low-cost sensors yield usable data when calibrated against research-grade reference instruments, although some drawbacks have also been reported. 

Optical particle sensors rely on elastic light scattering to measure time- and size-resolved PM concentrations; they are widely used in aerosol research, particularly when measuring particles in the 0.5 μm to 10 μm range. Aerosol photometers that simultaneously measure the bulk light scatter of multiple particles have limited success in laboratory studies [22]. Additionally, low-cost sensor measurements may suffer from sensor-to-sensor variability due to a lack of quality control and differences between individual components [22,23]. Sensor geometry can be optimized to reduce the effect of the particle Complex Refraction Index (CRI). CRI sensitivity can be addressed by designs that measure scattered light at multiple different angles simultaneously [24] or by employing dual-wavelength techniques [25]. However, these solutions involve complex and expensive components not suitable for compact, low-cost devices. Optimizing the detector angle relative to the excitation beam can reduce dependency on CRI [26] in the compact form; however, this approach has not been translated to high-volume fabrication. 

Calibration of the Plantower PMS A003 sensor (Plantower, Beijing Ereach Technology Co., Ltd., Beijing, China) used in the study was previously performed against the Tapered Element Oscillating Microbalance (TEOM), and against the Beta Attenuation Monitor (BAM) and Federal Reference Method (FRM) measurements [21,27]. The integrated mass measurements cannot account for temporal particle size and concentration variation that can occur during the calibration experiment. The instruments that directly measure aerosol size and concentration in real-time can be a better fit for sensor calibration [22,28].

Time and space-resolved 3D aerosol monitoring requires compact instruments to provide accurate particle sizing and cost-effective operational reliability [29]. These instruments must provide data to the user in real-time, enabling corrective action-based temporal and geospatial analysis. The traditional methods require laboratory-based testing methods, which cannot achieve real-time analysis. The system used in this study, AeroSpec (shown in Figure 2), consists of compact area monitors and a wearable Personal Exposure Monitor (PEM) that incorporates a PMS sensor, humidity and temperature sensors, and Wi-Fi and cellular Long Term Evolution (LTE) chips for data transfer [30]. The data can be transmitted in real-time to the database using Wi-Fi or a cellular connection. The Secure Digital (SD) card is used for on-board data backup. The real-time sensor data is aggregated to create a 3D map of size-resolved aerosol concentration. Figure 3 demonstrates the flow diagram of the AeroSpec system and how the sensors’ data-driven intervention approach can alert users of aerosol transmission routes.

The current methodologies have limitations in analyzing space-resolved aerosol distribution during clinical procedures. Past studies typically utilized settling plates or filter paper to collect aerosol and splatter samples [31,32]. These studies also used dental simulation units, as opposed to real patients. A study conducted at the School of Dental Sciences in Newcastle performed a crown preparation procedure on a dental simulation unit for the upper-right central incisor. A full-mouth scaling procedure was done using a magnetostrictive ultrasonic scaler [33]. The methodology consisted of placing filter paper around the immediate area and investigating the aerosol distribution and transmission routes using photographic and spectrofluorometric analyses. It was found that the observed aerosol and splatter concentration was the highest within one and a half meters of the origin, with some aerosol readings documented as far as four meters away. The study reported that when a handheld dental suction unit is incorporated, aerosols’ contamination is reduced by 75% within 0.5 and 1.5 m of the origin.

The University of North Carolina at Chapel Hill conducted a study that evaluated the use of an Extraoral Suction Unit (ESU) [34]. A dental manikin was used in a restorative treatment procedure. A color-changing polyvinyl chloride electrical tape grid was placed around the manikin’s mouth to detect any splatter. Photo-analysis of this grid was used to determine splatter concentration and variation. Without ESU-use, 34% of the experimental grid’s total surface area was found to have droplet splatter. When the ESU was set to level four out of 10 speed levels, it fell to 16%, and later 8% when the ESU was set to level 10. The ESU was then set to level 10, and droplet splatter coverage was documented to have reduced to 8%. This study’s findings concluded that handheld suction units are more effective in lowering aerosol distribution than the standing unit vacuums.

A study conducted by the Lublin University of Technology used Optical Particle Sizing (OPS) units, placed inside and outside of a dental office over nine days [35]. It was found that the particle concentrations reached the highest increase during dental grinding and drilling procedures. However, there were no reports on the effectiveness of any intervention strategies used during the dental procedure.

In this paper, the AeroSpec sensor network is presented as a new methodology to measure particle concentration in dental facilities. The tests were performed at the University of Washington’s School of Dentistry. The sensor network recorded time and space-resolved particulate matter concentration in real-time and produced a 3D map of particle distribution. We evaluated two most common intervention strategies, the local HEPA filtration unit and EOSD device. A network of 14 sensors was used to evaluate the EOSD device’s effectiveness during high-aerosolizing procedures (restoration and cleaning); a network of three sensors was also used to evaluate the efficiency of the HEPA filtration unit. Finally, the results from both HEPA filters and EOSD intervention strategies are outlined, along with a discussion on how to use the AeroSpec network to mitigate aerosol transmission in a dental setting.

## 2. Materials and Methods

### 2.1. Experimental Setup

AeroSpec sensors were used to evaluate the effectiveness of space filtration units, as well as extra-oral suction devices. Table 1 includes the naming of all devices and equipment used in the study. 

#### 2.1.1. Evaluating Aera Filtration Units

In assessing the area filtration units’ efficiency, sensors were placed near a Surgically Clean Air Jade air purifier to measure the particle concentration the devices emitted. Three sensors were placed in the immediate vicinity of an isolated purifying unit in the main hallway of UW SoD. Sensor AS8 was hung approximately 60 cm above the air purifying unit’s outlet fan, while sensor AS3 was placed on the floor next to the unit. The final sensor AS13 was placed on top of a trash bin next to the air purifying unit. The trash bin was taller than the air purifying unit, resulting in the sensor being approximately one foot higher than the air-purifying unit’s outlet fan. Figure 4 shows the sensor network set up for evaluating the HEPA filtration unit.

#### 2.1.2. Evaluating EOSD Units

To evaluate the persistence of dental aerosols during the operation and effectiveness of EOSD, 14 AeroSpec devices were placed in an operatory located in a high-traffic educational dental clinic at different heights (shown in Figure 5). Four of the devices were placed in the hallway located between operatories C8 and D8 (shown in Figure 6). The sensor labeled AS4 was attached close to the dentist’s breathing area to measure the exact amount of personal exposure associated with different activities and dental procedures.

Dentists performed regular restoration and cleaning activities while the sensor network gathered data, as shown in Figure 6. All particles greater than 0.3 µm in size were captured by the sensors and displayed every 10 s in real-time. Table 2 presents a list of events during the procedure that could have affected the area of interest. The two most effective events were Event Number Two (i.e., when the dentist began the tooth-drilling portion of the procedure) and Event Number Four (i.e., when the EOSD machine was turned to high in cubicle D8). The data collected during the cleaning procedure in cubicle C8 have been omitted due to the short window of aerosol generation available during the monitoring session.

## 3. Results and Discussion

The results gathered from analyzing the HEPA filter are shown in Figure 7. While the sensor placed on the floor (AS3) and the nearby trashcan (AS13) show consistent results, a drop of approximately 500 in particulate count of Dp>0.3 is observed by the sensor placed next to the unit (AS8), where following the fan when turned to low speed, particle concentration doubles. Alternatively, a drop in concentration is witnessed when the fan is placed on high, and an even further reduction is observed once the fan is placed on max speed. However, the overall particle concentration reduction is minimal, caused by the unit’s fan circulating the surrounding air. Upon further observation of the Surgically Clean Air Jade air purifier [36], it was noticed that the HEPA filter was not correctly sealed, causing particles to escape. As the fan’s flow rate starts to increase, the particle concentration begins to drop in response due to a higher rate of dissipation.

Figure 8 highlights the particle concentration data measured by AS4, AS3, and AS1 devices. The dental personnel wore sensor AS4. Sensor AS3 was placed closest to the patient’s mouth, and AS1 was placed in the hallway. The goal of the analysis is to evaluate the effectiveness of the existing intervention strategies during the COVID-19 pandemic. The main objective was to look for high exposures around the patient and physician’s breathing zones (the peaks shown in red circles in Figure 7), as well as the potential of contaminating surrounding operatories. The results demonstrated that the EOSD units are somewhat effective around the patient’s mouth and the dental personnel’s breathing zone (reduced hazardous peaks once EOSD is turned on). 

To calculate the total change in particle concentration before and after running the EOSD, the total number of particles produced was calculated (shown in (1)) from sensors AS1 (hallway), AS3 (patient), and AS4 (physician): (1)$$P_{D}=\overset{t_{f}}{\underset{t=t_{i}}{\sum }}P_{t}$$
where $P_{D}$ is the total number of particles produced by the drilling procedure, *t_i_* is the time when the drilling procedure started (2:12 PM), and *t_f_* is when the procedure ended (2:35 PM). Sensor AS1 lowered to an average of 357 particles, with sensors AS3 and AS4 reporting an average of 409 and 394 particles. This shows a percentage decrease of 16.4%, 12%, and 21.5% for AS1, AS3, and AS4, respectively. The total average reduction was 16.6% in particle concentration after approximately five minutes of EOSD use:(2)$$d=\frac{n_{A}-n_{B}}{n_{B}}\times 100$$
where *d* represents the percentage decrease, with $n_{B}$ and $n_{A}$ representing the concentration before and after the EOSD was implemented, respectively, in (2).

The data gathered while investigating the EOSD unit showed that the drilling procedures produced an average of 465 particles with sensors AS1 (hallway), AS3 (patient), and AS4 (physician), recording an average of 427, 465, and 503 particles, respectively. For the five minutes the EOSD was run, the average particle concentration decreased to around 387 particles.

While the average concentration of aerosols was low in the observed dental operatories, personnel-based PM sensors did document high levels of individual exposure, demonstrating the need for updated intervention strategies to negate the effects of possibly harmful aerosols. Figure 10 illustrates a 3D visualization of evaluated particulate distributions at varying levels in dental practice. The addition of this visualization tactic would support new intervention strategies, such as adopting external air filtration systems at their respective optimal heights.

## 4. Future Work

Further research will demonstrate how AeroSpec sensors monitor aerosols’ generation and evaluate the effectiveness of intervention strategies used during dental procedures. This study’s limitation was the short sampling time, creating a need to conduct aerosol monitoring for a more extended period of time within the dental setting. Recording a more extended period of EOSD and air purifying units is suggested to further validate them as an existing mitigation technique. Analyzing for dead zones—areas where particles may get trapped—and areas such as waiting rooms or hallways to be monitored for high concentration can help indicate if additional areas need intervention strategies. Future studies, alongside this one, will provide the dental community with the tools necessary to determine a standardized intervention strategy to address infectious aerosol persistence in dental clinics.

## 5. Conclusions

The objective of this study was to evaluate the effectiveness of existing aerosol distribution intervention strategies using AeroSpec monitoring systems. Current approaches applied by dental clinics lack evidence to support their efficacy in reducing aerosols. The methodology presented in this paper provides real-time aerosol distribution in 3D, which is used to determine effective interventions in aerosol evacuation during dental procedures.

The data collected from this study found a slight reduction in particle count when EOSD units were turned on. However, the study showed that dental personnel are at the highest risk when performing high-aerosolizing procedures. In addition, an increased number of particle concentrations traveled across the hallway into other operatories, putting other patients and dental personnel at risk.

With the restrictions imposed due to the ongoing COVID-19 pandemic, the inaccuracy and drift tendencies of currently available methodologies make evaluating the efficiency of mitigation techniques challenging. Understanding how dental fields can reduce the spread of particles from high aerosol-generating procedures will further protect technicians and clients of any given facility.

## Figures and Tables

**Figure 1 sensors-21-03928-f001:**
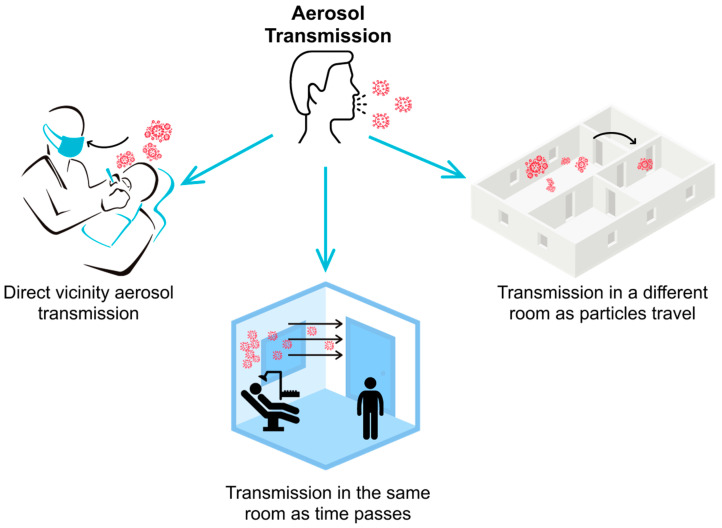
Aerosolized particles generated during dental operations can transmit infectious diseases via three paths: (**a**) direct-vicinity transmission (dental professionals while performing high aerosol-generating procedures); (**b**) same-room transmission as time passes (anyone who enters the room after the procedure); and (**c**) different room or location transmission (aerosolized particles can travel to different rooms or locations in the dental office, putting everyone in those locations at risk).

**Figure 2 sensors-21-03928-f002:**
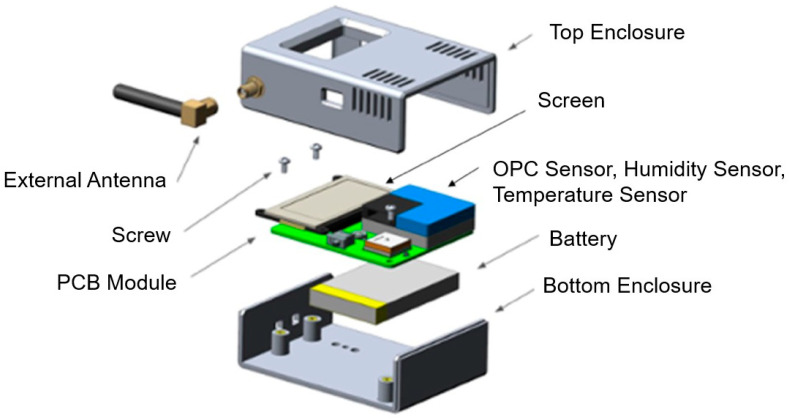
Exploded view of the AeroSpec hardware unit, which consists of environmental sensors, the PCB board, LiPo battery, and screen. The electronics were assembled in an ABS plastic enclosure manufactured by Tool Less Plastics. The device dimensions are (H) 100 mm × (W) 60 mm × (D) 25 mm and weighs 120 g.

**Figure 3 sensors-21-03928-f003:**
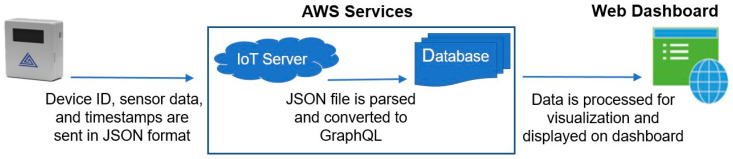
The AeroSpec sensor network system diagram. The hardware unit transmits the data to the AWS database as JSON format, and the analyzed data are displayed in 3D visualization format on the web-based software in real-time using the AWS database.

**Figure 4 sensors-21-03928-f004:**
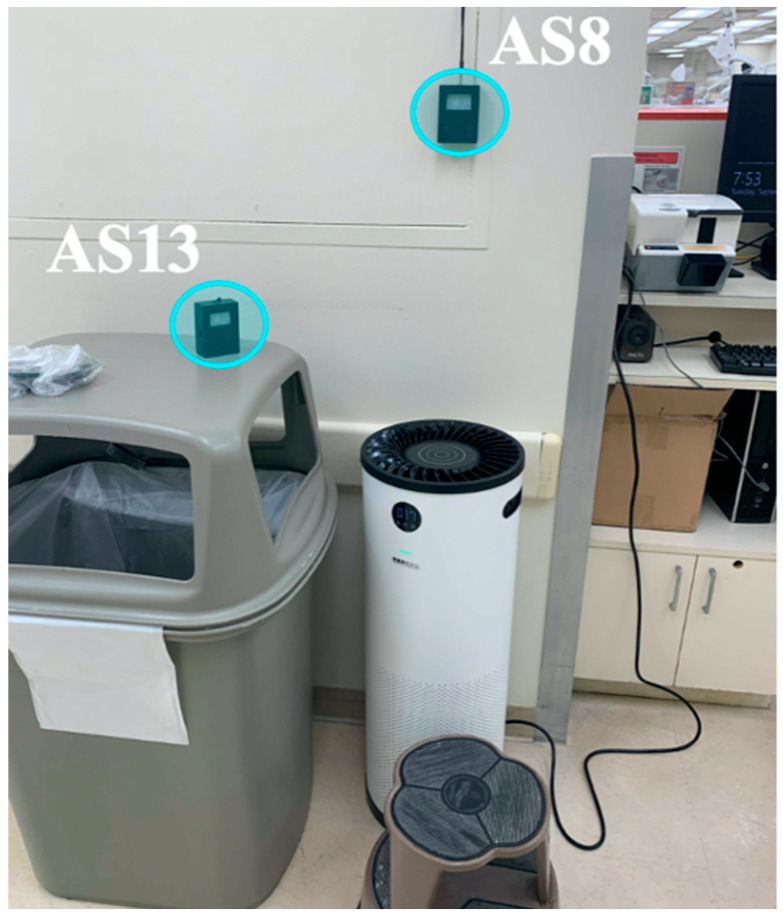
Photograph of the Surgically Clean Air Jade air purifier with three sensors (one sensor not in the field of view, located on the floor, in front of the air inlet of the filtration unit) to monitor the effects of the HEPA-filter-equipped air purifier.

**Figure 5 sensors-21-03928-f005:**
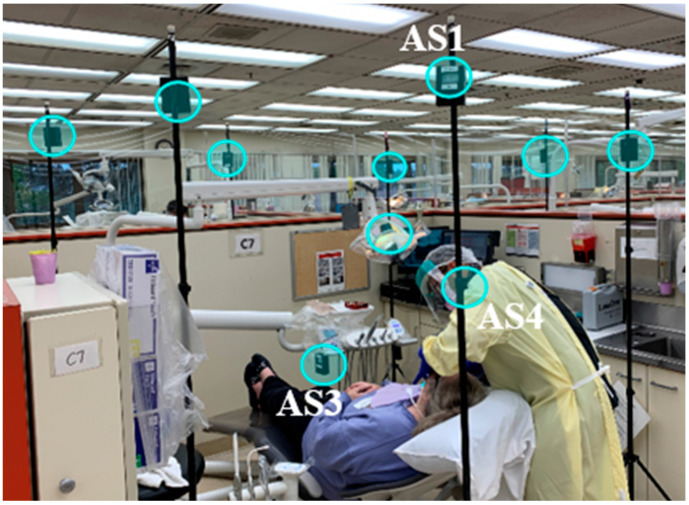
Photograph of the dental facility with 14 sensors (some sensors not in the field of camera view) to monitor dentists’ and patients’ exposure during drilling, composite removal, and cleaning operations. Sensors were placed around the operatory, as well as the surrounding bays, at two different heights.

**Figure 6 sensors-21-03928-f006:**
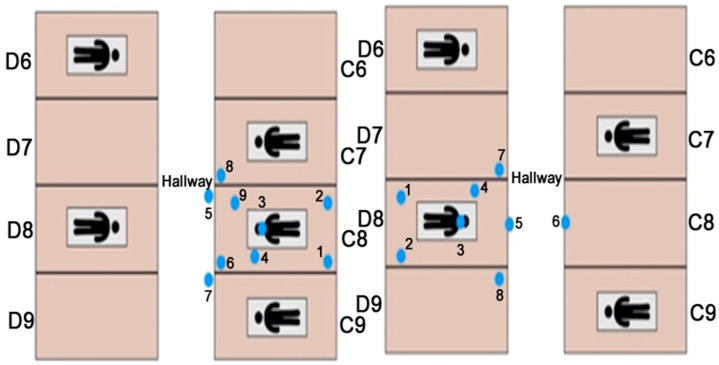
Positions of the sensors’ network were tested in a dental operatory consisting of 16 individual bays. Two bays, C8 and D8, were equipped with the sensors.

**Figure 7 sensors-21-03928-f007:**
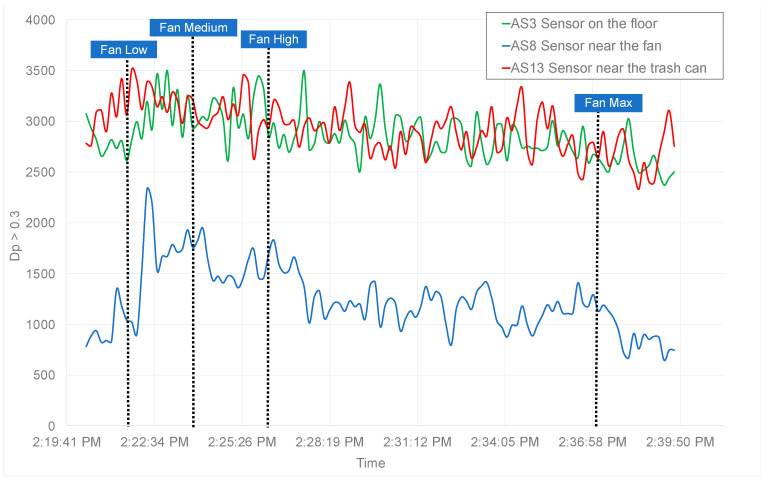
The number of particulates larger than PM 0.3 (Dp > 0.3) data from three sensors AS3, AS8, and AS13 used to evaluate the effectiveness of the Surgically Clean Air Jade HEPA air purifier is presented. The data show a minimal change in particulate count regardless of the fan speed. After further investigation, it was discovered that the HEPA filter was not sealed properly, which caused the particles to escape and get circulated by the fan.

**Figure 8 sensors-21-03928-f008:**
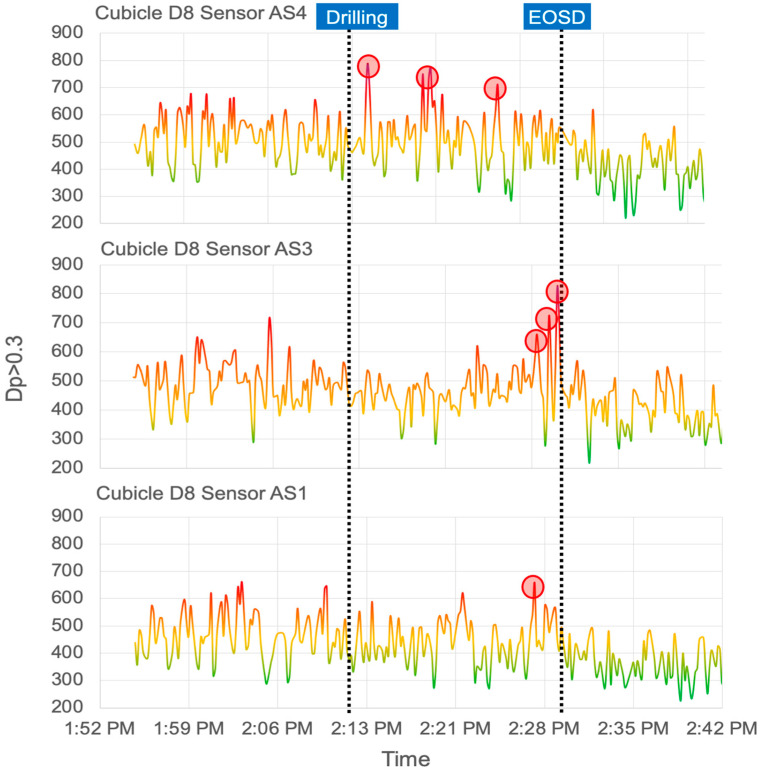
The number of particulates larger than PM 0.3 (Dp > 0.3) data from three selected sensors is presented. The high exposure peaks are shown in red circles. AS4 was worn by the physician, AS3 was placed closest to the patient, and AS1 was placed in the hallway outside the dental operatory bay. The spikes associated with the drilling procedure are immediately detected by AS4, putting the physician at the highest risk of transmission. Some spikes are detected by AS1, which shows that aerosolized particles are traveling across the hall and potentially contaminating other locations. The graphs have been color-coded in five categories, which are defined in Figure 9.

**Figure 9 sensors-21-03928-f009:**
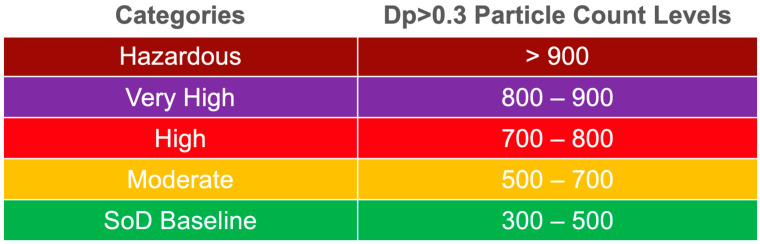
Five categories were chosen to represent different levels of exposure rates. The SoD baseline was calculated separately during active hours. The categories were defined using the same structure as PM 2.5 published by EPA [37].

**Figure 10 sensors-21-03928-f010:**
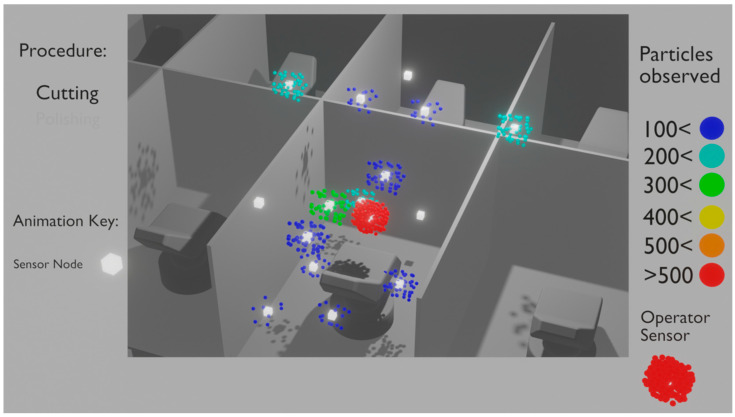
Real-time 3D visualization of the dental operatory D8 during a drilling procedure. Due to its importance, the operator’s device is highlighted in the bottom-right corner of the screen. While the aerosol concentration is at a safe level inside the cubicle, the physician is being exposed to a very high concentration of aerosols. Please see Appendix A.

**Table 1 sensors-21-03928-t001:** Explanation of the devices and variables used in the study.

Devices and Terminologies	Definitions
AS1, AS2, etc.	AeroSpec (AS) sensor number
EOSD	Extra-Oral Suction Device used during procedures
Surgically Clean Air Jade	Space filtration units used continuously
C8, D8	Dental operatory room numbers

**Table 2 sensors-21-03928-t002:** Procedures in Cubicle D8.

Event #	Time	Description
#1	2:01 PM	Cubicle D8 restoration procedure begins; Cubicle C7 has impression procedure underway.
**#2 (Drilling)**	**2:12 PM**	**Cubicle D8 drilling procedure begins.**
#3	2:19 PM	Cubicle C9 is cutting procedure underway.
#3a	2:31 PM	Lost connection (B17).
**#4 (EOSD ON)**	**2:35 PM**	**Suction machine #1 turned on in cubicle D8 (on high mode).**
#5	2:37 PM	Cubicle C8 started the examination.
#6	2:40 PM	Suction machine #1 turned off.
#6a	2:41 PM	Reset Sensor (B17).

## Data Availability

The data presented in this study are available on request from the corresponding author. The data are not publicly available due to privacy.

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
