# Peer review of "Methodology for Addressing Infectious Aerosol Persistence in Real-Time Using Sensor Network"

_sensors, 2021, doi:10.3390/s21113928_

Round 1
Reviewer 1 Report
Your study was well designed to collect data with smart sensor network in the dental setting and then analyse the aerosol particle distribution. The weakness of this paper is less data to strong support your study. I suggest authors to look at all data collected by 14 sensors to further consolidate your study. I also suggest authors to check the line 17 area-filtration or air-filtration? same words appear in the article.
Author Response
Point 1: "I suggest authors to look at all data collected by 14 sensors to further consolidate your study."
The authors would like to thank the reviewer for their feedback and suggestions. We agree that the data presented in this manuscript is not comprehensive and requires further investigation to standardize the effectiveness of the current intervention strategies. Having that said, this study was meant to evaluate current techniques used to reduce the risk of COVID-19 transmission. We are currently working with Professor Chan to do an in-depth investigation on extra-oral suction devices, which we will report the results in our future manuscript.
We have adjusted the graph and added new text to clarify this concern.
Reviewer 2 Report
Manuscript review for “Methodology for Addressing Infectious Aerosol Persistence in Real-Time Using Sensor Network”
- Comments to the authors –
Topic Scope
The authors present a new methodology through a combination of real-time sensors and 3D visualization can provide dental personnel and facility mangers with actionable feedback to effectively assess aerosol transmission in medical settings and develop evidence-based intervention strategies.
The work presented here is strongly connected to the field of sensors and fits well to the broad frame provided by Sensors. This fact is also reflected by the bibliographic references used in this manuscript, which are -in the majority- closely related to the field of sensors.
I support publication in Sensors, as the authors demonstrate that the combination of real-time measurements and 3D visualization could improve estrategies to mitigate the aerosol transmistion. However, I have a question about the treatment followed with the complex refractive index of aerosol. The elastic scattering estructure rely on the particle size and the real and the imaginary refractive indexes. If particles are transparents, then, to retrieve particle size from elastic scattering (phase functions) is direct, but if the aerosol is absorbent the imaginary refractive index is relevant and the elastic scattering curve is smoothed. Therefore, to estimate particle size is more challenge. I would like the authors clarified my doubt about this aspect and provide detailed information about the treatment followed to consider the imaginary refractive index of aerosol.
Author Response
Point 1: I have a question about the treatment followed with the complex refractive index of aerosol. The elastic scattering estructure rely on the particle size and the real and the imaginary refractive indexes. If particles are transparents, then, to retrieve particle size from elastic scattering (phase functions) is direct, but if the aerosol is absorbent the imaginary refractive index is relevant and the elastic scattering curve is smoothed. Therefore, to estimate particle size is more challenge. I would like the authors clarified my doubt about this aspect and provide detailed information about the treatment followed to consider the imaginary refractive index of aerosol.
NaCl particle were used in the study. We agree that the data reported by the sensor is subject to errors. Calibration of the sensors for NaCl is beyond the scope of this paper and OEM calibration was used. We have added the following text to give background related to low-cost optical sensors there dependency on CRI and calibration.
Low-cost particulate matter (PM) sensors find increasing use in various applications, including monitoring AQ in the outdoor [1-4] and indoor environment [5]. Time-resolved exposure data from wearable monitors [6] can be used to assess individual exposure in near real-time. As low-cost sensors find applications in pollution monitoring, there is a need to characterize and calibrate their performance under various conditions, and especially in controlled environments with standardized test aerosols. Various studies have evaluated the performance of low-cost PM sensors in laboratory and field settings [7-14]; these reports show that low-cost sensors yield usable data when calibrated against research-grade reference instruments, although some drawbacks have also been reported.
Optical particle sensors rely on elastic light scattering to measure time- and size-resolved PM concentrations; they are widely used in aerosol research, particularly when measuring particles in the 0.5 μm to 10 μm range. Aerosol photometers that measure the bulk light scatter of multiple particles simultaneously have limited success in laboratory studies [14]. Also, low-cost sensor measurements may suffer from sensor-to-sensor variability due to a lack of quality control and differences between individual components [14, 15]. Sensor geometry can be optimized to reduce the effect of particle complex index of refraction (CRI). CRI sensitivity can be addressed by designs that measure scattered light at multiple different angles simultaneously [16] or by employing dual-wavelength techniques [17]. However, these solutions involve complex and expensive components not suitable for compact, low-cost devices. Optimizing the detector angle relative to the excitation beam can reduce dependency on CRI [18] in the compact form; however, this approach has not been translated to high volume fabrication.
Calibration of the Plantower PMS A003 sensor (Plantower, Beijing Ereach Technology Co., Ltd, China) used in the study was previously performed against the tapered element oscillating microbalance (TEOM), and against the beta attenuation monitor (BAM) and federal reference method (FRM) measurements [13, 19]. The integrated mass measurements cannot account for temporal particle size and concentration variation that can occur during the calibration experiment. The instruments that directly measure aerosol size and concentration in real-time can be a better fit for sensor calibration [14, 20].
Reviewer 3 Report
The manuscript is well written and very easy to understand. I have found only two issues:
* Line 260: there is no connection between this line and the formula presented bellow.
* Section 2 – Materials and Methods: in my opinion, this section needs an introductory text in order to define the environment and the used equipment. For example, you mentioned AS8, AS13, … during the text, however you do not define these terms.
The main drawback of the manuscript is the experiments. You have measured the environment only once. You have to perform the measurement several times to provide better statistical data. Furthermore, I suggest the use of trend lines in the graphs presented by Figures 7 and 8. It can help readers to see the tendency of the presented data.
Author Response
The authors would like to thank the reviewer for their feedback and suggestions. Please see our responses below:
Point 1: Line 260: there is no connection between this line and the formula presented bellow.
This concern has been addressed in line 255 to 257
Point2: Section 2 – Materials and Methods: in my opinion, this section needs an introductory text in order to define the environment and the used equipment. For example, you mentioned AS8, AS13, … during the text, however you do not define these terms.
This concern has been addressed and shown in Table I.
Point 3: The main drawback of the manuscript is the experiments. You have measured the environment only once. You have to perform the measurement several times to provide better statistical data. Furthermore, I suggest the use of trend lines in the graphs presented by Figures 7 and 8. It can help readers to see the tendency of the presented data.
The authors would like to thank the reviewer for their feedback and suggestions. We agree that the data presented in this manuscript is not comprehensive and requires further investigation to standardize the effectiveness of the current intervention strategies. Having that said, this study was meant to evaluate current techniques used to reduce the risk of COVID-19 transmission. We are currently working with Professor Chan to do an in-depth investigation on extra-oral suction devices, which we will report the results in our future manuscript.
We have adjusted the graph and added new text to clarify this concern.